# The Influence of Smoking Status on Exhaled Breath Profiles in Asthma and COPD Patients

**DOI:** 10.3390/molecules26051357

**Published:** 2021-03-04

**Authors:** Stefania Principe, Job J.M.H. van Bragt, Cristina Longo, Rianne de Vries, Peter J. Sterk, Nicola Scichilone, Susanne J.H. Vijverberg, Anke H. Maitland-van der Zee

**Affiliations:** 1Department of Respiratory Medicine, Amsterdam UMC, University of Amsterdam, 1105 AZ Amsterdam, The Netherlands; stefaniaprincipe90@gmail.com (S.P.); j.j.vanbragt@amsterdamumc.nl (J.J.M.H.v.B.); c.longo@amsterdamumc.nl (C.L.); riannedevries1@gmail.com (R.d.V.); p.j.sterk@amsterdamumc.nl (P.J.S.); s.j.vijverberg@amsterdamumc.nl (S.J.H.V.); 2Dipartimento Universitario di Promozione della Salute, Materno Infantile, University of Palermo, Medicina Interna e Specialistica di Eccellenza “G. D’Alessandro”(PROMISE) c/o Pneumologia, AOUP “Policlinico Paolo Giaccone”, 90127 Palermo, Italy; nicola.scichilone@unipa.it; 3Breathomix b.v., 2333 Leiden, The Netherlands

**Keywords:** exhaled breath, eNose, smoking, asthma, COPD

## Abstract

Breath analysis using eNose technology can be used to discriminate between asthma and COPD patients, but it remains unclear whether results are influenced by smoking status. We aim to study whether eNose can discriminate between ever- vs. never-smokers and smoking <24 vs. >24 h before the exhaled breath, and if smoking can be considered a confounder that influences eNose results. We performed a cross-sectional analysis in adults with asthma or chronic obstructive pulmonary disease (COPD), and healthy controls. Ever-smokers were defined as patients with current or past smoking habits. eNose measurements were performed by using the SpiroNose. The principal component (PC) described the eNose signals, and linear discriminant analysis determined if PCs classified ever-smokers vs. never-smokers and smoking <24 vs. >24 h. The area under the receiver–operator characteristic curve (AUC) assessed the accuracy of the models. We selected 593 ever-smokers (167 smoked <24 h before measurement) and 303 never-smokers and measured the exhaled breath profiles of discriminated ever- and never-smokers (AUC: 0.74; 95% CI: 0.66–0.81), and no cigarette consumption <24h (AUC 0.54, 95% CI: 0.43–0.65). In healthy controls, the eNose did not discriminate between ever or never-smokers (AUC 0.54; 95% CI: 0.49–0.60) and recent cigarette consumption (AUC 0.60; 95% CI: 0.50–0.69). The eNose could distinguish between ever and never-smokers in asthma and COPD patients, but not recent smokers. Recent smoking is not a confounding factor of eNose breath profiles.

## 1. Introduction

Asthma and chronic obstructive pulmonary disease (COPD) are complex and heterogeneous chronic airway diseases that include several phenotypes, characterized by different inflammatory pathways [1,2]. The complexity and the heterogeneity of these diseases is due to variability of clinical characteristics, environmental influences, and pathophysiology aspects that are different for each patient [3]. Therefore, there is still a clinical need for new biomarkers to characterize the underlying processes [4].

The study of exhaled breath composition (“breathomics”), could facilitate a phenotyping approach of chronic airway diseases [5]. Exhaled breath is partially composed of volatile organic compounds (VOCs), including exogenous VOCs (e.g., drinks, food, drugs, environment) and endogenous VOCs (e.g., microbiome and body (patho) physiological metabolic processes), which can directly originate from the metabolism of bacteria residing in alveoli, immune cells, etc., and can also diffuse into the bloodstream where they diffuse passively across the capillary/alveolar interface and are subsequently emanated in the exhaled breath with different configurations according to their origin [6,7,8]. The eNose, a non-invasive and rapid technique that is able to detect exhaled VOC patterns, has shown some promise in characterizing asthma and COPD based on inflammatory characteristics and discriminating between patients with asthma, COPD, and lung cancer [9,10].

Exhaled breath measurements are considered a promising diagnostic technique, being easy to perform and potentially giving additional information that may help phenotyping patients, there are still some limitations related to the fact that there are many factors (e.g., diet, smoking, co-morbidities, physical activities, age, gender, pregnancy, and medication use) which could influence the level of individual compounds present in the exhaled breath [11]. Since VOCs are the result of metabolic and inflammatory processes related to (patho) physiological changes that take place in the respiratory tract [12], and smoking contributes to altering these processes in asthma and COPD [13], it is critical to assess the sensitivity of the eNose for the smoking status of patients with chronic respiratory diseases. Therefore, we aimed to investigate whether the eNose is suitable as a non-invasive technique to identify how patients with different smoking habits may respond to smoke exposure and whether smoking has an influence on disease classification. To assess this, in this exploratory analysis, we hypothesized that the eNose is able to distinguish between ever- and never-smokers.

Among patients with asthma and COPD and healthy volunteers, we assessed whether the eNose can accurately discriminate between (1) ever- vs. never-smokers, and (2) smoking less than vs. greater than 24 h before the exhaled breath measurement.

## 2. Results

### 2.1. Baseline Characteristics and Study Design

The study subjects selected for the analysis were enrolled from December 2015 through May 2017 across six different sites. Of the included asthma and COPD patients (n = 896) 593 (60.4%) were ever-smokers (237 asthma and 356 COPD) and 303 (33.8%) were never-smokers (295 asthma and 8 COPD). Among the ever-smokers, 167 (28.2%) smoked their last cigarette <24 h before measurement. The healthy control group was composed of 199 ever-smokers (out of which 107 subjects smoked in the last 24 h) and 366 never smokers. Table 1, Table 2 and Table 3 display the baseline clinical (age, body mass index (BMI), gender, medications) and functional (FEV1, FVC, FEV1/FVC) characteristics of the overall population (Table 1), ever-smokers (Table 2), and the healthy controls (Table 3). Ever-smokers had more advanced age and worse lung function (FEV1/ FEV1/FVC). In the asthma and COPD training set, 469 patients were ever-smokers and 247 were never-smokers. In the validation set, 124 patients were ever-smokers and 56 were never-smokers. A flowchart of the study design is presented in Figure 1a. Among the healthy controls, 452 patients were in the training set (160 ever-smokers and 292 never-smokers), and 113 subjects were in the validation set (39 ever-smokers and 74 never-smokers) (Figure 1b).

### 2.2. The Ability of the eNose to Discriminate a History of Smoking in Asthma and COPD Patients

Out of 13 sensors, three principal components (PCs) were selected that captured 64% of the variance within the dataset of asthma and COPD patients (PC1 37%, PC2 16%, PC3 9%). There was no significant correlation between relevant PCs and ever- or never-smoker patients with asthma or COPD (see Appendix A). The ability to classify a history of smoking in patients with asthma or COPD showed reasonable accuracy in the training set (area under the receiver–operator characteristic curve (ROC–AUC) = 0.74, 95% CI = 0.70–0.77), and this accuracy was further confirmed in the validation set (ROC–AUC = 0.75, 95% CI = 0.68–0.82), with 67% of cross-validated grouped cases correctly classified after Leave One Out Cross-Validation (LOO-CV) in both groups (Figure 2a). This was confirmed with a higher accuracy using the number of pack-years among ever-smoker patients in both the training and validation sets (see Appendix A).

### 2.3. The Ability of the eNose to Discriminate Recent Cigarette Consumption in Asthma and COPD Patients

The eNose could less accurately identify patients with recent last cigarette consumption (<24 h) compared to smoking patients with a cigarette consumption >24 h in both the training and validation sets (ROC–AUC = 0.60; 95% CI = 0.54–0.65; ROC–AUC= 0.55; 95% CI = 0.44–0.67 respectively). eNose was not able to discriminate who smoked their last cigarette before and after 24 h before the visit (Figure 2b) with an accuracy of 51% after LOO-CV in both sets.

### 2.4. Does Smoking Influence eNose Results?

The same analysis was repeated with the healthy control group. Three PCs (out of 13 sensors) were selected that captured 61% of the variance within the dataset (PC1 34%, PC2 14%, PC3 13%). The eNose was not able to distinguish among ever- and never-smokers in either the training (ROC–AUC: 0.54; 95% CI: 0.49–0.60) or the validation sets (ROC–AUC: 0.56; 95% CI: 0.50–0.69) with an accuracy of 53% after LOO-CV in both groups (Figure 3a). This was confirmed according to the number of pack/years among ever-smokers (see Appendix A). Moreover, the eNose was not able to discriminate between subjects who smoked their last cigarette shorter or longer than 24 h prior to the exhaled breath measurement in both the training and the validation sets (training: area under curve (AUC): 0.60; 95% CI: 0.50–0.69; validation: AUC: 0.60; 95% CI: 0.47–0.70) (Figure 3b).

These results were further confirmed when repeating the analysis sub setting among ever-smoker healthy subjects, current smokers, and ex-smokers. The eNose was not accurate enough to distinguish between ex- (n = 139) and never-smokers (n = 366) in the training or the validation sets (training set AUC: 0.52; 95% CI: 0.46–0.58; validation set AUC: 0.56; 95% CI: 0.46–0.60) (Figure 3c). The eNose did not distinguish between ever-, current- (n = 60), and never-smokers (n = 366) in the training set (AUC: 0.62; 95% CI: 0.51–0.67) or the validation set (AUC: 0.65; 95% CI: 0.53–0.71) (Figure 3d).

Interestingly, the eNose was not able to distinguish between ever- and never-smokers in the asthma group (see Appendix A), but it could discriminate the diagnosis of asthma and COPD patients among the ever-smoker population (see Appendix A).

## 3. Discussion

In this study, we demonstrated that the use of the eNose to analyze exhaled breath can discriminate patients with a chronic respiratory disease (asthma or COPD) with and without a smoking history, but we could not make a distinction between smokers that did or did not smoke a cigarette in the last 24 h. We also demonstrated that the eNose is not influenced by smoking history in healthy volunteers; therefore, we can assume smoking may not be considered as a confounder that interferes with eNose measurements. These results were internally validated and were confirmed in an independent validation set.

Exhaled breath measurement by an eNose device has been used as a non-invasive tool for detecting several diseases with screening and diagnostic implications [10,14,15,16]. To our knowledge, this is the first study that evaluates the ability of the eNose to discriminate patients with chronic airway diseases according to their smoking status and the possible detection of the influence of smoking in both pathologies.

We can assume that the association of exhaled breath and eNose signals most likely reflects pathophysiological modifications related to the underlying chronic airway disease and combined airway alterations due to chronic smoking exposure. These results are in line with other studies. In a dual-center study [17] that recruited 222 smokers and non-smokers, with or without COPD, the eNose was able to classify COPD never- and ex-smokers and COPD active-smokers. Interestingly, a proportion of current smokers (9.3%) was misclassified non-smokers according to the analysis of their CO levels, which seems to confirm that, in line with our results, the eNose can distinguish among patients with chronic smoking habits, whereas it is not able to detect smoking in patients depending on the time of last cigarette consumption. Also, Papaefstathiou et al. [18] recently demonstrated that exhaled breath can be used to discriminate between smokers, non-smokers, and e-cigarette users in a population of healthy subjects and, in particular, that relevant VOCs can be identified among these three groups. Moreover, the diagnostic accuracy of exhaled breath analysis, linked to routine spirometry for chronic airway diseases, was previously assessed by De Vries et al. [10]. eNose patterns were found to be predictive for the differential diagnosis of asthma and COPD (AUC 0.88) and, in line with our results, eNose breath profiles did not show any ability to discriminate between current and ex-smokers (AUC 0.52) among patients with COPD, even though ROC analyses showed high accuracy in detecting exacerbations with the population stratified for pack-years [19]. Compared to our results, we are able to moderately distinguish between ever- and never-smokers (AUC 0.74), and we conclude that smoking could play a role in the VOCs contributing to the ability of the eNose to distinguish between asthma and COPD patients, but the eNose likely also detects other pathological factors that may be characteristic for these chronic respiratory diseases.

Furthermore, VOCs have been previously used to discriminate different inflammatory patterns in several chronic respiratory diseases, enabling researchers to obtain subgroups based on molecular characteristics [20,21,22]. Recently, Caruso et al. [23] used metabolomic analysis of VOCs in exhaled breath to identify a “severe asthma smoking phenotype,” showing that, in line with our results, the analysis of VOCs identified differences among severe asthmatic smokers and ex-smokers, compared to never smokers. However, the severe asthmatics with smoking history were almost all ex-smokers, meaning that the differences may not have been related to active smoking, but potentially to the eNose results reflecting damage in the airways caused by smoking in the past [24].

We also considered whether smoking is a confounder of eNose results with respect to chronic respiratory disease phenotypes. We therefore also included data on the healthy controls. Smoking could influence the levels of individual compounds present in the exhaled breath and could therefore hamper the implementation of exhaled breath analysis as a diagnostic tool [25]. To our knowledge, this is the first study that demonstrates that recent or past smoking does not influence breath patterns in healthy subjects, in line with the hypothesis that breath patterns most likely reflect airway alterations caused by past smoking.

The strengths of our study are the relatively large sample size, the BreathCloud cohort, which recruited patients from different centers, obtaining a mixed population resembling the general COPD and asthma population, and the use of standardized methods for the analysis, including the internal and external validations that were performed to support the obtained results. The limitations of our study were that there was no information collected related to passive smoking that may have indirectly influenced never smokers and we had no information related to urinary nicotine concentration, which could be a more accurate measure, but more burdensome for patients [26]. A further limitation of our study is that the eNose can identify patterns of VOC mixture rather than the individual compounds that are driving the signal, even though this characteristic makes eNose breath profiles as suitable as composite multidimensional biomarkers in providing numerical probabilities for the presence or absence of a particular clinical condition [27,28]. On the other hand, the advantages are that this technology is noninvasive, easy to use, and results can be promptly available and interpretable for clinicians.

## 4. Materials and Methods

### 4.1. Study Design

We conducted a cross-sectional analysis using exhaled breath and clinical information obtained from the multicenter BreathCloud [9] database. BreathCloud enrolled patients with asthma, COPD, lung cancer, cystic fibrosis, and healthy volunteers from ten different centers in the Netherlands during routine outpatient visits. The following data from medical records collected general characteristics (age, BMI, gender, allergy history) symptoms assessment (asthma control questionnaire (ACQ), clinical COPD questionnaire (CCQ), oral corticosteroid assumption), functional tests (e.g., spirometry pre- and post- bronchodilator) and, among the ever-smokers, whether they smoked before and after 24 h. The exhaled breath measurements were collected in routine clinical practice and were subsequently handled in compliance with the Dutch Personal Data Protection Act (WPB).

### 4.2. Subject Selection

Patients and healthy controls were enrolled by six centers of primary, secondary and tertiary care in the Netherlands. Patients were included in this analysis if they were ≥18 years old, had a physician-reported diagnosis of asthma or COPD, and had answered the questions about smoking history. Healthy subjects were those who did not report any history of asthma or COPD, and who did not use any respiratory medications. Patients were stratified according to their smoking habits. Patients with a recent history of acute upper or lower respiratory tract infections were excluded because a history of upper [29,30,31] or lower [32,33,34] respiratory infections may influence the quality of breath samplings, and we did not know how much this could interfere with the resulting breath pattern profiles in patients with a diagnosis of asthma and COPD. The purpose of adding the SpiroNose to routine diagnostics was explained to the patients, who all gave their oral consent before enrollment. Due to the non-invasive nature of the BreathCloud study, the Amsterdam UMC medical ethical review board provided a waiver for ethical approval of the protocol (reference: W14_112#14.17.0147). All six sites made use of the same sampling protocol, which was part of the AMC MRB approval no: 14.17.0147.

### 4.3. Smoking Definitions

Patient-reported smoking history was chosen as an outcome, according to previous studies [35,36] that demonstrated that self-reported smoking is accurate. Smoking status was further divided into ever- and never-smokers; ever-smokers were considered active smokers who currently smoke cigarettes (number of pack/year) and former smokers who had smoked at least 100 cigarettes in their lifetime, but have quit smoking. Never-smokers were patients without any history of smoking habits, or who had smoked fewer than 100 cigarettes in their lifetime. The number of pack-years was calculated as (number of cigarettes smoked per day/20) × number of years smoked.

The second smoking definition was patient-reported recent cigarette smoking, which was defined as having smoked a cigarette in the 24 h prior to the exhaled breath measurement.

### 4.4. Exhaled Breath Measurements

Exhaled breath samples were collected using the SpiroNose [10]; an eNose composed of seven separate cross-reactive metal oxide semiconductor (MOS) sensors used to detect exhaled breath VOCs while monitoring for ambient VOCs. The SpiroNose comprises 7 different MOS sensors, each present in duplicate in both the reference and the exhaled breath sensor arrays. The MOS sensors (Figaro Engineering Inc., Osaka, Japan) were chosen based on their good stability and long-term performance [37]. MOS sensors operate with temperatures ranging between −40 °C and +70 °C. Using thick film techniques, the sensor material was printed onto electrodes on an alumina substrate. Tin dioxide (SnO_2_) was the main sensing material of the sensor element [38]. From each sensor signal, two variables were derived. The SpiroNose provided a spectrum of signals representing 13 data points originated by 6 sensor peaks normalized to sensor 2, the most stable sensor, and 7 peak/breath hold ratios. Each SpiroNose sensor signal had a high sensitivity to different mixtures of volatile organic compounds (VOCs)/gases in the exhaled breath and the reference (ambient air) sensor arrays [39].

Before breath measurement, patients had to rinse their mouth thoroughly with water three times, and then perform five tidal breaths, after which they maximally inhaled and held their breath for 5 s before slowly exhaling. The measurement was performed two times, with an interval of two minutes between maneuvers. Data were sent in real-time and stored on the online BreathCloud server.

### 4.5. Statistical Analysis

A descriptive analysis was performed to generate tables with the general characteristics of the population (Table 1, Table 2 and Table 3, Appendix A). A chi-squared test was used for categorical variables and a one-way ANOVA test was used for continuous variables. A principal component (PC) analysis was performed to summarize the eNose breath signals. According to the Kaiser criterion [40], all PCs with an eigenvalue >1 were considered for the analysis. PCs were constructed for the overall number of subjects (including both the training and validation sets). Furthermore, a linear discriminant analysis (LDA) was used to determine whether PCs could accurately classify patient-reported smoking history (ever-smokers vs. never-smokers) and, among ever-smokers, recent cigarette consumption (<24 h vs. >24 h). Internal validation was performed with leave-one-out cross-validation and by a split analysis in which the LDA model constructed with the training set was applied to the validation set. The area under the receiver–operator characteristic curve (ROC) was used to assess the accuracy of the models and it was obtained from the prediction made by the LDA model. The dataset was randomly divided into a training set containing 80% of the data (total asthma and COPD group = 716; asthma = 426; COPD = 288; healthy controls = 452) and a validation set including 20% of the data (total asthma and COPD group = 180; asthma = 106; COPD = 74; healthy controls = 113). The model acquired with the training set was used to retrieve similar variables in the validation set. A sensitivity analysis was also performed, using the number of pack-years of the ever-smokers (see Appendix A). Supplementary analyses concerning only the asthma group are reported in the Appendix A; an additional analysis assessing the accuracy of the eNose in distinguishing asthma and COPD among a population of ever-smokers is also reported in Appendix A. Data selected had no missing sensor values in BreathCloud.

The analysis was performed using R studio version 1.1.463 (R Studio Inc., Boston, MA, USA) and using R version 3.5.1 (The R Foundation for Statistical Computing, Vienna, Austria), with packages; dplyr, caret, pROC, and MASS [41,42].

## 5. Conclusions

We demonstrated that a smoking history might influence eNose breath profiles in patients with chronic airway diseases, while we cannot distinguish patients and healthy subjects according to recent cigarette consumption. This means that we can measure the influence of smoking on airways, but not the cigarette smoke itself. The present findings are in support of the usage of eNose technology as a quick and feasible technique for the diagnosis and phenotyping of chronic airway diseases in a clinical setting.

## Figures and Tables

**Figure 1 molecules-26-01357-f001:**
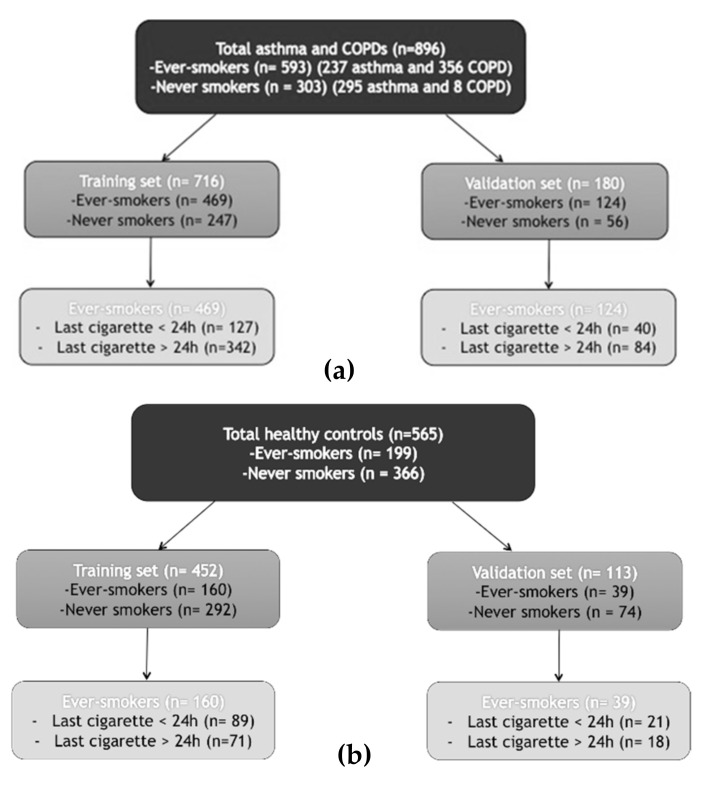
Flowchart of the study design of the asthma and COPD group (**a**), and healthy controls (**b**).

**Figure 2 molecules-26-01357-f002:**
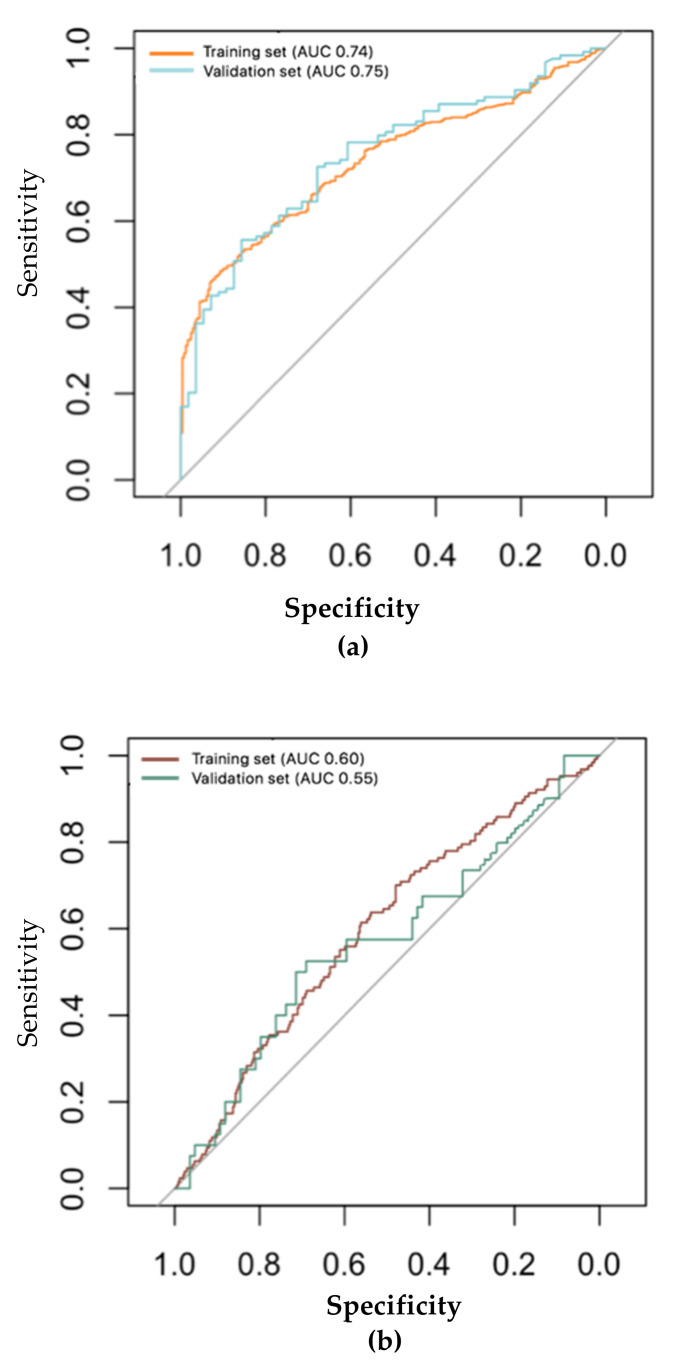
ROC analyses showing the accuracy of the linear discriminant model based on principal component (PC) reduction in the training set and the independent validation set for the asthma and COPD group. (**a**) Ever-smokers with asthma and COPD: training set (n = 469) 95% CI: 0.70–0.77 area under curve (AUC): 0.74; validation set (n = 124) 95% CI: 0.68–0.82 AUC: 0.75. (**b**) Time of last cigarette assumption in asthma and COPD patients (control: more than 24 h; case: less than 24 h): training set case = 127; control = 583; 95% CI: 0.54–0.65; AUC: 0.60; validation set case = 40; control = 84; 95% CI: 0.44–0.67; AUC: 0.55.

**Figure 3 molecules-26-01357-f003:**
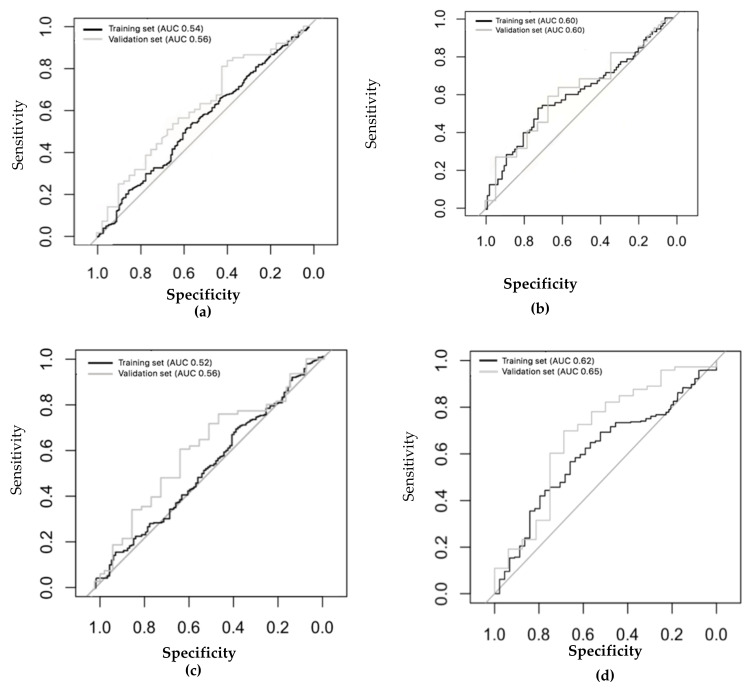
ROC analyses showing the accuracy of the linear discriminant model based on principal component reduction in the training set and the independent validation set for the healthy control group. (**a**) Ever-smokers in healthy subjects: training set (n = 452) AUC: 0.54 (95% CI: 0.49–0.60); validation set (n = 113) AUC: 0.60 (95% CI: 0.50–0.69). (**b**) Time of last cigarette consumption in healthy subjects (control: more than 24h; case: less than 24h) training set (n = 160) AUC: 0.60 (95% CI: 0.50–0.69); validation set (n = 39) AUC: 0.60 (95% CI: 0.42–0.70). (**c**) Ex- (n = 139) and never-smokers (n = 366): training set (ex-smokers = 116; never-smokers = 292), AUC: 0.52 (95% CI: 0.46–0.58); validation set (ex-smokers = 23; never-smokers = 74) AUC: 0.56 (95% CI: 0.46–0.60). (**d**) Active (n = 60) and never smokers (n = 366): training set (active smokers = 44; never-smokers = 292) AUC: 0.62 (95% CI: 0.51–0.67); validation set (active smokers = 16; never smokers = 74) AUC: 0.65 (95% CI: 0.53–0.71).

**Table 1 molecules-26-01357-t001:** Demographics of asthma and chronic obstructive pulmonary disease (COPD) patients stratified by smoking status. Data are expressed in number of patients, mean ± standard deviation or median and range for non-normal distributions.

Asthma and COPD Patients	Ever Smokers (n = 593)	Never Smokers (n = 303)	*p*-Value
Age (mean (SD))	60.99 (13.38)	48.46 (18.04)	<0.001
BMI (mean (SD))	27.78 (5.82)	26.95 (6.69)	0.055
Gender = M/F (%)	48.6/51.4	37.3/62.7	0.002
Allergy = Yes/No (%)	42.0/58.0	69.6/30.4	<0.001
FEV (mean (SD)) (l)	2.04 (0.89)	2.62 (0.93)	<0.001
FVC (mean (SD)) (l)	3.41 (1.07)	3.66 (1.13)	0.002
FEV1/FVC (mean (SD)) (%)	56 (16)	70 (14)	<0.001
FEV1 pred (mean (SD)) (%)	70.93 (23.81)	86.14 (21.74)	<0.001
ACQ (median [IQR])	1.60 [0.86, 2.50]	1.43 [0.71, 2.29]	0.208
CCQ (median [IQR])	1.00 [0.00, 2.30]	0.00 [0.00, 0.00]	<0.001
Current use of ICS = No/Yes (%)	27.8/72.2	15.5/84.5	<0.001
Oral corticosteroids (%)			0.780
Current use	2.4	1.7	
Previous use	10.3	10.6	
No	87.4	87.8	

BMI: body mass index; ACQ: asthma control questionnaire; CCQ: clinical COPD questionnaire; ICS: inhaled corticosteroids.

**Table 2 molecules-26-01357-t002:** Clinical characteristics of patients with recent cigarette consumption. Data are expressed in number of patients, mean ± standard deviation or median and range for non-normal distributions.

Ever Smokers (n = 593)	<24 h (n = 167)	>24 h (n = 426)	*p*-Value
Age (mean (SD))	56.98 (14.97)	60.83 (13.57)	<0.001
BMI (mean (SD))	26.63 (5.76)	27.75 (5.80)	0.114
Gender = M/F (%)	47.9/52.1	27.7/72.3	0.001
FEV1 (mean (SD)) (l)	2.03 (0.87)	2.05 (0.90)	<0.001
FVC (mean (SD)) (l)	3.37 (1.05)	3.41 (1.08)	<0.001
FEV1/FVC (mean (SD)) (%)	56 (16)	56 (16)	<0.001
FEV1 pred (mean (SD)) (%)	68.82 (21.79)	70.99 (23.83)	<0.001
Pack/year (median [IQR])	30.00 [17.00, 48.00]	25.00 [10.95, 41.25]	<0.001
ACQ (median [IQR])	1.86 [1.14, 2.86]	1.60 [0.88, 2.50]	0.010
CCQ (median [IQR])	1.40 [0.00, 2.55]	1.00 [0.00, 2.30]	<0.001

BMI: body mass index; ACQ: asthma control questionnaire; CCQ: clinical COPD questionnaire.

**Table 3 molecules-26-01357-t003:** Demographics of healthy subjects stratified by smoking status. Data are expressed in number of patients, mean ± standard deviation.

	Ever Smokers (n = 199)	Never Smokers (n = 366)	*p*-Value
Age (mean (SD))	46.95 (15.29)	35.67 (14.05)	<0.001
BMI (mean (SD))	26.02 (4.86)	23.72 (3.65)	<0.001
Gender = M/F (%)	72/127 (36.2/63.8)	127/239 (34.7/ 65.3)	0.795
FEV1(%) (mean (SD))	89.17(12.43)	92.08 (15.39)	<0.001
FEV1/VC (%) (mean (SD))	91.96 (12.26)	94.88(14.53)	<0.001
Last cigarette (%)			<0.001
<24 h	107 (53.7)	0	
>24 h	92 (46.2)	0	
Pack/years (mean (SD))	15.77 (19.20)	0	<0.001

## Data Availability

The data presented in this study are available on request from the corresponding author. The data are not publicly available due to privacy.

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
