# Peer review of "The Influence of Smoking Status on Exhaled Breath Profiles in Asthma and COPD Patients"

_molecules, 2021, doi:10.3390/molecules26051357_

Round 1
Reviewer 1 Report
This study tried to explore the influence of smoking on exhaled breath profiles in asthma and COPD patients, whose exhalations were detected by an e-nose. The study population is large and it is interesting to see a partial influence of smoking on the e-Nose breath profiles in patients with chronic airway diseases. This means that this e-nose based on the MOS sensor array can measure the influence of smoking on airways, but not the cigarette smoke itself. However, the main problem is the methodology is not clear. What were the MOS sensors that composed the e-nose? How many variables were extracted from these 7 sensors, and what were they? What were the univariate and multivariate methods used for the statistical analysis? Was the ROC obtained from the prediction by the LDA-model? Were these analyses adjusted by age, gender, and BMI, since these confounders were not balanced in the study?
Other comments:
#1. Since the subjects were selected from December 2015 until May 2017 across 6 different sites and no clinical trial was registered, did all the centers used the same protocols for the breath collection?
#2. Why Patients with a recent history of acute upper or lower respiratory tract infections were excluded?
#3. This could have better to list asthma and COPD separately in demographics: age, gender, allergy, etc. of ever smokers than never smokers to better understand the influence of smoking.
#4. Did you perform linear discriminant analysis (LDA) separately on asthma and COPD patients?
#5. Patients who present with a combination of asthma and COPD related traits are not uncommon, were there any patients with the same traits present in your data?
#6. Fig. 2 and 3 were not clear, and the title of the x-axis should be specificity but not 1-specificity, otherwise, the value of the x-axis should be 0 to 1 but not 1 to 0. The same was presented in the supplementary figures.
Reviewer 2 Report
The subject is very interesting, and it is one to which the authors have made significant contribution. The experimental section is explained well in details and the results are understandable for the readers. However, the quality of presentation is not acceptable for a scientific publication.
Overall, the manuscript is acceptably written in an engaging style with an appropriate level to our readership. The paper is suitable for publication after improving the quality of the presented figures.
Author Response
We thank the reviewer for the comment that allowed us to improve the quality of the paper. All the figures have been modified as suggested in the revised Manuscript and the Supplementary material.
Reviewer 3 Report
Today, the field of diagnosing diseases using exhaled air is very promising. The manuscript is devoted to the study of the effect of smoking on the composition of exhaled air in the diagnosis of asthma and COPD. The authors obtained very interesting and valuable scientific results using eNose technology. This work can be recommended to publication after minor improvements.
- I recommend using the following order of parts. Introduction - Materials and methods - Results - Discussion - Conclusion.
- For research, the authors used a SpiroNose sensor. In order to facilitate the use of the obtained data by other scientific groups and, in particular, to improve the Enose sensors, it is necessary to add more information on the characteristics of the metal oxide semiconductor sensors that were used.
